# Direct formation of HONO through aqueous-phase photolysis of organic nitrates

Juan Miguel González-Sánchez[1,2], Miquel Huix-Rotllant[2], Nicolas Brun[1,2], Julien Morin[1], Carine Demelas[1], Amandine Durand[1], Sylvain Ravier[1], Jean-Louis Clément[2], Anne Monod[1]

[1]Aix Marseille Univ, CNRS, LCE, Marseille, France
[2]Aix Marseille Univ, CNRS, ICR, Marseille, France

*Correspondence to*: Juan Miguel González-Sánchez (juangonzalez.sc@proton.me) and Anne Monod (anne.monod@univ-amu.fr)

**Abstract.**

Organic nitrates ($RONO_2$) are secondary compounds whose fate is closely related to the transport and removal of $NO_x$ in the atmosphere. Despite their ubiquitous presence in submicron aerosols, the photochemistry of $RONO_2$ has only been investigated in the gas phase, leaving their reactivity in condensed phases poorly explored. This work aims to address this gap by investigating, for the first time, the reaction products, and the mechanisms of aqueous-phase photolysis of four $RONO_2$ (i.e., isopropyl nitrate, isobutyl nitrate, α-nitrooxyacetone, and 1-nitrooxy-2-propanol). The results show that the reactivity of $RONO_2$ in the aqueous phase differs significantly from that in the gas phase. In contrast to the gas phase, where $RONO_2$ releases $NO_x$ upon photolysis, the aqueous phase photolysis of $RONO_2$ leads primarily to the direct formation of HONO, which was confirmed by quantum chemistry calculations. Hence, the aqueous-phase photolysis of $RONO_2$ represents both a $NO_x$ sink and a source of atmospheric nitrous acid (HONO or $HNO_2$), a significant precursor of ·OH and ·NO. These secondary radicals (·OH and ·NO) are efficiently trapped in the aqueous phase, leading to the formation of $HNO_3$ and functionalized $RONO_2$. This reactivity can thus potentially contribute to the aging of Secondary Organic Aerosol (SOA) and serve as an additional source of aqueous-phase SOA.

## 1 Introduction

Organic nitrates are secondary compounds, formed through $NO_x$ + VOC reactions, that play an essential role in the transport and removal of $NO_x$ in the atmosphere. These compounds can have long lifetimes, lasting from a few hours to several days, which allow them to travel to remote regions (Shepson, 1999). During their long-range transport, they can undergo reactions such as gas-phase photolysis and/or ·OH oxidation, which can release $NO_x$ back into the atmosphere. As a result, $RONO_2$ molecules are responsible for a more uniform distribution of $NO_x$ and thus, are indirectly responsible for the transport of other pollutants such as $O_3$ and SOA (Perring et al., 2013).

Furthermore, RONO$_2$ also participate in NO$_x$ removal from the atmosphere can occur either through their deposition to the Earth's surface or by transformation into a less reactive chemical compound, such as nitric acid (Hu et al., 2011; Nguyen et al., 2015). Therefore, their atmospheric reactivity and fate must be considered to accurately predict pollution transport on a regional scale. This is especially important for world regions experiencing decreasing NO$_x$ levels, such as Europe and North America, where the relative importance of RONO$_2$ in NO$_x$ transport and removal is growing due to the increase of the overall

transformation of NO$_x$ into RONO$_2$ (Romer Present et al., 2020).

RONO$_2$ are not only present in the gas phase, as some of them have low volatility and can partition into condensed phases. As a result, RONO$_2$ account for a significant fraction of submicron organic aerosol, ranging from 5% to 77% (Kiendler-Scharr et al., 2016; Ng et al., 2017). RONO$_2$ reactivity in condensed phases may differ from that in the gas phase and may affect their role as NO$_x$ reservoirs. For instance, it is well-established that the hydrolysis of tertiary and allylic RONO$_2$ serves as a fast and

permanent sink of NO$_x$ in the atmosphere, as the nitrate group is transformed into nitric acid (Darer et al., 2011; Rindelaub et al., 2015; Hu et al., 2011). However, only a small fraction of RONO$_2$ (between 9 % and 34 % for α- and β-pinene related RONO$_2$) undergoes hydrolysis (Takeuchi and Ng, 2018; Wang et al., 2021). Other aqueous-phase reactions are thus to be considered: photolysis and ·OH oxidation. Our previous studies have emphasized the significance of aqueous-phase reactivity for atmospherically relevant RONO$_2$, such as isoprene and terpene nitrates, with intermediate to high water solubilities

(González-Sánchez et al., 2021, 2023). At typical cloud/fog conditions (liquid water content, LWC, of 0.35 g m$^{-3}$), the aqueous-phase photoreactivity can act as a major sink (> 50 %) for water-soluble RONO$_2$ (K$_H$ > 10$^5$ M atm$^{-1}$), while at very low LWC (3 ·10$^{-5}$ g m$^{-3}$), it can serve as a major sink for very highly water-soluble RONO$_2$ (K$_H$ > 10$^9$ M atm$^{-1}$). Nevertheless, the fate of the nitrate group during these processes is still unknown, and it is uncertain whether this reactivity acts as a NO$_x$ sink or as an additional transport mechanism.

This work intends to address these questions for the aqueous-phase photolysis of RONO$_2$. While aqueous-phase photolysis is a minor sink when compared to ·OH oxidation (González-Sánchez et al., 2023), the investigation of the photolysis mechanisms is of high importance as it is a first step towards the fundamental understanding of RONO$_2$ photooxidation pathways.

To explore the aqueous-phase photolysis of RONO$_2$, the fate of four molecules (i.e., isopropyl nitrate, isobutyl nitrate, α-nitrooxyacetone, and 1-nitrooxy-2-propanol) was experimentally investigated. These RONO$_2$ served as proxies to understand

the fate of the nitrate group. The two alkyl nitrates are simple molecules, simplifying the comprehension of mechanisms related to nitrate group reactivity. Furthermore, the other two investigated RONO$_2$ are polyfunctional, they combine highly relevant functional groups (hydroxy and carbonyl groups) with the nitrate group, allowing for the assessment of their influence on the reactivity. The aqueous-phase photolysis primary and secondary reaction products were identified and quantified, and the fate of the nitrate group was elucidated with support from theoretical calculations. The atmospheric implications of these findings

are discussed.

## 2 Materials and methods

### 2.1 Experimental setup

The aqueous-phase photolysis experiments were conducted using the experimental setup previously described in detail by González-Sánchez et al., (2023). Briefly, a 450 $cm^3$ double-wall Pyrex aqueous-phase photoreactor filled with 400 mL of aqueous solution and covered by a quartz lid was used. It was equipped with four apertures for reagent injections, sampling, and an Optical IDS dissolved oxygen sensor FDO® 925 (WTW) which included temperature monitoring. The reactor was thermostated at 298 K and continuously stirred. Irradiation was provided by an arc light source (LOT Quantum Design) equipped with a 1000 W arc Xe lamp. Wavelengths below 290 nm were removed using an ASTM 892 AM1.5 standard filter (see lamp spectra in Fig. S1 along with the liquid-phase absorption cross-sections of the investigated $RONO_2$). González-Sánchez et al., 2023A constant distance of 18.4 cm between the lamp and the water surface was carefully maintained in all experiments.

### 2.2 Photolysis experiments

Prior to each photolysis experiment, the photoreactor was filled with mili-Q water and the $RONO_2$ was added. The solution was stirred for 30 min in the dark to ensure complete dilution of the $RONO_2$. Meanwhile, the lamp was turned on for 10 min to stabilize the light beam. The first aliquot was sampled when the reactor was placed under the light beam, marking the reaction time zero. Photolysis reactions were performed for 4 to 7 h at 298.0 ± 0.2 K. The specific experimental conditions of all photolysis experiments are appended in Table 1.

During the reaction, aliquots were regularly sampled for offline analyses. The pH of the reaction mixture was measured using a 9110DJWP pH Probe (Thermo Scientific). Ultra-High Performance Liquid Chromatography-Ultra-Violet detector (UHPLC-UV) analyses were performed to monitor the $RONO_2$ decay or to identify and quantify carbonyl compounds after 2,4-dinitrophenylhydrazine (DNPH) derivatization. High-Performance Ionic Chromatography-Conductivity Detector (HPIC-CD) analyses were conducted to quantify $HNO_2$, $HNO_3$ and organic acids. At the end of the reaction, the remaining volume was used to perform liquid-liquid extraction and Gas Chromatograph-Mass Spectrometer (GC-MS) analyses to identify the formed oxidized $RONO_2$. In experiment 1, the headspace of the reactor was monitored with a $NO_x$ analyzer to investigate the possible formation of these compounds.

**Table 1: Initial conditions and analytical instruments used during the photolysis experiments of 4 individual $RONO_2$ molecules.**

| Nº | $RONO_2$ | $[RONO_2]_0$ / mM | Reaction time / h | UHPLC-UV | DNPH[a] | HPIC | GC-MS | $NO_x$ Analyzer |
|----|----------|-------------------|-------------------|----------|---------|------|-------|-----------------|
| 1 | Isopropyl nitrate | 1.00 | 4 | | | | | X |
| 2 | Isopropyl nitrate | 0.93 | 7 | X | X | X | X | |
| 3 | Isopropyl nitrate | 1.81 | 7 | X | X | | | |
| 4 | Isopropyl nitrate | 1.71 | 5 | X | X | X | | |

| | | | | | | | |
|---|---|---|---|---|---|---|---|
| 5 | Isobutyl nitrate | 0.60 | 7 | X | | | |
| 6 | Isobutyl nitrate | 0.59 | 7 | X | | | |
| 7 | Isobutyl nitrate | 0.53 | 7 | X | | | | X |
| 8 | Isobutyl nitrate | 0.55 | 7 | X | X | X | |
| 9 | Isobutyl nitrate | 0.49 | 7 | X | X | | |
| 10 | α-Nitrooxyacetone | 1.18 | 7 | X | X | X | X |
| 11 | 1-Nitrooxy-2-propanol | 0.72 | 7 | X | X | X | X |
| 12 | 1-Nitrooxy-2-propanol | 0.38 | 7 | X | X | X | |

**ᵃUHPLC-UV analyses after DNPH derivatization of the sample were used to identify and quantify carbonyl compounds. All experiments were performed at 298.0 ± 0.2 K.**


### 2.3 Analytical instruments

### 2.3.1 UHPLC-UV

An Ultra-High Performance Liquid Chromatography (UHPLC) with an Ultra-Violet detector (UV) (Thermo Scientific Accela) equipped with a Hypersil Gold C18 column (50 x 2.1 mm) with a particle size of 1.9 µm and an injection loop of 5 µL was

used to quantify carbonyl compounds after 2,4-dinitrophenylhydrazine (DNPH) derivatization and $RONO_2$.

**a) Measurements of $RONO_2$.**

A binary eluent of $H_2O$ and $CH_3CN$ was used for all analyses, with a flow rate of 400 µL min$^{-1}$. Two gradients were used depending on the polarity of the compounds. For isopropyl nitrate and isobutyl nitrate, the gradient started at $H_2O/CH_3CN$

80/20 (v/v) and was gradually adjusted to 50/50 (v/v) over 3 min, held at this proportion for 1 min, and then set back to 80/20 (v/v) within 10 s until the end of the run at 5 min. For more polar compounds, i.e., α-nitrooxyacetone and 1-nitrooxy-2-propanol, a similar gradient was employed but the initial and final proportions were adjusted to $H_2O/CH_3CN$ 90/10 (v/v) to optimize their retention times (rt). All $RONO_2$ were detected at their maximum absorbance wavelength at 200 nm (González-Sánchez et al., 2023).

Calibration curves were linear (as determined by the Mandel test) between $5 \cdot 10^{-5}$ and $1 \cdot 10^{-3}$ mol L$^{-1}$ with $R^2 > 0.9995$. Aliquots with expected concentrations higher than $1 \cdot 10^{-3}$ mol L$^{-1}$ were diluted before analyses. The retention times were 0.9, 1.2, 2.4, and 3.3 min for 1-nitrooxy-2-propanol, α-nitrooxyacetone, isopropyl nitrate, and isobutyl nitrate, respectively (Figure S2). Limits of detection (LOD) were $9 \cdot 10^{-6}$ mol L$^{-1}$ for isopropyl nitrate and $1 \cdot 10^{-5}$ mol L$^{-1}$ for the 3 other compounds.

**b) Measurements of carbonyl compounds.**

To derivatize the carbonyl compounds, 500 µL of the aqueous sample was mixed with 450 µL of 0.005 M DNPH and 50 µL of 0.1 M HCl. The mixture was allowed to react for 24 hours to achieve high yields. A specific method was developed for separating and quantifying the formed hydrazones. The gradient, with a flow rate of 400 µL min$^{-1}$, started from $H_2O/CH_3CN$

80/20 (v/v) for 1 min, then was gradually adjusted to 30/70 (v/v) over 6 min, held at this proportion for 1 min, and then set back to 80/20 (v/v) within 10 s until the end of the run, at 9 min.

The resulting hydrazones from formaldehyde, acetaldehyde, acetone, hydroxyacetone, and isobutyraldehyde were identified and quantified at 360 nm. External calibrations were performed to quantify carbonyl compounds with concentrations ranging from $5 \cdot 10^{-6}$ to $1 \cdot 10^{-3}$ M and $R^2 > 0.9995$. Their retention times were 4.4, 5.1, 5.8, 3.8, and 6.8 min, respectively. LOD were 4.1, 2.1, 1.5, 4.2, and $5.3 \cdot 10^{-6}$ M, respectively.

### 2.3.2 HPIC-CD

The formation of $HNO_2$ and $HNO_3$ and organic acids such as formic acid and acetic acid was quantified using a DIONEX ICS-3000 High-Performance Ionic Chromatography (HPIC) with a DIONEX IonPac™ AG11-HC precolumn (4 x 50 mm) and a DIONEX IonPac™ AS11-HC column (4 x 250 mm) coupled to a CD25 conductivity detector.

A binary eluent gradient method composed of $H_2O$ and NaOH 0.1 mol $L^{-1}$ aqueous solution was optimized to separate the formed organic acids at relatively short retention times. At a flow rate of 1 mL $min^{-1}$, the gradient started at $H_2O$/NaOH 0.1 mol $L^{-1}$ 96/4 (v/v) for 10 min, then gradually to 50/50 (v/v) during 12 min, then went back within a minute to 96/4 (v/v), and was held at this proportion until the end of the analyses at 25 min. The injection volume was 200 µL, and a constant flow of $H_2SO_4$ 0.05 M continuously passed through the suppressor at a flow rate of 3 mL $min^{-1}$.

The retention times of acetate, formate, $NO_2^-$ and $NO_3^-$ were 5.9, 7.2, 17.3, and 21.9 min, respectively. Calibration curves were optimized to obtain good linearity and low LOD (within the concentrations range expected). LOD were $4.3 \cdot 10^{-6}$, $3.5 \cdot 10^{-6}$, $6 \cdot 10^{-7}$ and $5 \cdot 10^{-7}$ M for acetic acid, formic acid, $NO_2^-$ and $NO_3^-$, respectively.

### 2.3.3 GC-MS

A Clarus® 680 Gas Chromatograph (GC, Perkin Elmer) equipped with an Elite-5MS Capillary Column (Perkin Elmer) with 30 m length, 0.25 mm diameter, and 0.25 µm of film thickness coupled to an AxION® iQT™ Quadrupole/Time of Flight-Mass Spectrometer (MS, Perkin Elmer) was used to qualitatively detect and identify oxidized $RONO_2$ formed during the aqueous-phase photolysis experiments. $RONO_2$ were extracted and preconcentrated from the remaining solution after the end of each photolysis experiment. The remaining solutions were stored at ~4 °C for up to 48 h before the analyses.

100 mL of the remaining solution were extracted using 3 x 20 mL of dichloromethane in a separatory funnel. UHPLC-UV analyses of the aqueous phase before and after the extraction confirmed that all $RONO_2$ efficiently partitioned to dichloromethane. The extracts were washed with 20 mL of mili-Q water and were concentrated in a TurboVap II system (Biotage). The concentration workstation used a nitrogen flow at 11 psi and a water bath at 30 °C to evaporate dichloromethane until a 500 µL sample was obtained.

One µL of the concentrated extract was then injected into the GC-MS. The carrier gas was helium at a flow rate of 1 mL $min^{-1}$. A split of 20:1 was used due to the high concentration of the compounds. The injector temperature was set to increase from

60 °C to 200 °C within 1 min to prevent RONO$_2$ thermolysis. The following program was set in the oven: 30 °C for 10 min; increase until 300 °C at a 15 °C min$^{-1}$ rate; and hold for 10 min at 300 °C before the end of the analyses.

The analytes were detected with a Time-of-Flight Mass Spectrometer using electron impact ionization with an electron energy of 70 eV and an ion source temperature of 250 °C. The ion source was turned on 5 – 7 minutes after the analysis started, to avoid the saturation of the source due to the solvent signal. The detector performed full scan measurements from m/z = 30 to

300 amu. The mass-to-charge ratio of the ion NO$_2^+$ (m/z = 46), specific to RONO$_2$, was extracted to detect these compounds. Seven known RONO$_2$ were analyzed by GC-MS to investigate their retention times and fragmentation patterns (Section S1).

### 2.3.4 NO$_x$ analyzer in the reactor's headspace

A CLD 88p Ecophysics NO$_x$ analyzer was used to determine if ·NO and ·NO$_2$ were formed and partitioned into the gas-phase headspace of the solution during the photolysis of isopropyl nitrate. Indeed, both ·NO and ·NO$_2$ are highly volatile compounds

($K_H$ = 1.8 ·10$^{-3}$ M atm$^{-1}$ and $K_H$ = 2.0 ·10$^{-2}$ M atm$^{-1}$, respectively, Sander, 2015) i.e. from 30 to 10$^7$ times more volatile than the investigated RONO$_2$. Therefore, if any ·NO or ·NO$_2$ were formed during the aqueous-phase photolysis, they would have partitioned to the reactor's headspace.

As the NO$_x$ analyzer monitored the headspace of the reactor, a specific experimental setup consisting of a hermetic one-liter three-neck round-bottom flask was used (Fig. S3). It was irradiated by the lamplight beam on its side. Note that since the

reactor's headspace was also illuminated, photolysis of isopropyl nitrate could occur in the reactor's headspace. However, although isopropyl nitrate is highly volatile, most of the compound remained in the aqueous phase in the time scale of the experiments (only 3 % of isopropyl nitrate partitioned into the reactor's headspace after 7 h, (González-Sánchez et al., 2023). The NO$_x$ Analyzer LOD is 0.1 ppbv for both ·NO and ·NO$_2$. Considering the gas phase dilutions performed downward the reactor, ·NO$_x$ could be detected, if formed, at concentrations higher than ~2 ppbv using this set-up. Although the CLD 88p

Ecophysics NO$_x$ analyzer uses a photolytic converter, interferences with the RONO$_2$ were observed. A slight proportion (less than 0.6 %) of gas-phase isopropyl nitrate was detected as ·NO$_2$. Further details are given in Section S2.

In addition, a control experiment was performed to test the efficiency of the gas-phase ·NO$_2$ photolysis and conversion to ·NO under our experimental conditions by bubbling gas-phase ·NO$_2$ into the reactor's aqueous phase and photolyzing it with the lamplight. The experimental setup is depicted in Fig. S4.

### 2.4 Molar yield determinations

The molar yields of the primary reaction products were determined by plotting their concentrations against $\Delta[RONO_2]$, that represents the consumption of the parent organic nitrate (Eq. 1).

$$yield~(\%) = \frac{[product]}{\Delta[RONO_2]} \cdot 100\%$$    (1)

Since the reaction products were susceptible to undergo photolysis over time, the yields were calculated for the initial aliquots, sampled during the first 1–2 h of reaction. The evaporation rate of some $RONO_2$ could be non-negligible compared to photolysis (González-Sánchez et al., 2023), $\Delta[RONO_2]$ was thus systematically corrected from evaporation using Eq. (2-3).

$$[RONO_2]_{reac} = [RONO_2]_0 e^{(-k_{exp}+k_{vap})\cdot t} \tag{2}$$

$$\Delta[RONO_2] = [RONO_2]_0 - [RONO_2]_{reac}, \tag{3}$$

where $[RONO_2]_0$ is the initial concentration, and $k_{exp}$ the pseudo-first-order decay, determined by fitting of the $RONO_2$ concentrations ($[RONO_2]_t$) versus time ($t$) following Eq. (4).

$$[RONO_2]_t = [RONO_2]_0 e^{(-k_{exp})\cdot t} \tag{4}$$

The evaporation rate constant, $k_{vap}$, was determined by control experiments reported in González-Sánchez et al., (2023).

## 2.5 Theoretical calculations

Theoretical simulations of the photolysis reaction of isopropyl nitrate photolysis were performed in a model of aqueous solution and in the gas phase. For building the model in the gas phase, snapshots from a 10 ps QM MD dynamics were performed using a thermostat at 300 K. For the static calculations of minima, conical intersections and transition states we used B3LYP/6-31G* for the ground state calculations and TDDFT using the Tamm-Dancoff approximation for the excited state calculations, with water treated as an implicit solvent via polarizable continuum model (PCM). To perform the quantum dynamics simulations, we built a model of isopropyl nitrate in aqueous solution, by constructing first a water box of (22.5 Angstroms)$^3$ equilibrated using Amber99 TIP3P water forcefield parameters. Isopropyl nitrate was then soaked in the water box, and re-equilibrated with the following protocol (see Section S3): 1) An NVT MM MD at fixed isopropyl geometry during 125 ps; 2) An NPT MM MD at fixed isopropyl during 1 ns; and 3) An NPT B3LYP/6-31G*//Amber99 QM/MM PBC MD relaxing the full system for 12 ps (Bonfrate et al., 2023). The snapshots were taken from the latter, discarding the first 2 ps. For each snapshot, a water droplet of 10 Angstroms was extracted, including a spherical wall potential to avoid evaporation of water during the excited-state dynamics. In each snapshot (gas phase and aqueous solution), non-adiabatic excited state molecular dynamics were operated using Tully's fewest switch surface hopping algorithm (Huix-Rotllant et al., 2023). The trajectories were started from the second excited state (S$_2$). Excited states were computed using mixed-reference time-dependent density-functional theory, which can describe the multi-configuration character of wavefunctions during photolysis at the cost of a density-functional theory calculation (Lee et al., 2018; Huix-Rotllant et al., 2023).

## 2.6 Reagents

Chemicals were commercially available and used as supplied: isopropyl nitrate (96%, Sigma Aldrich), isobutyl nitrate (98%, Sigma Aldrich), chloroacetone (95%, Sigma Aldrich), AgNO$_3$ (99%, VWR Chemicals), KI (98%, Sigma Aldrich), NaBH$_4$

(98%, Sigma Aldrich), 2,4-dinitrophenylhydrazine ($\geq$ 99 %, Sigma-Aldrich), formaldehyde-DNPH ($\geq$ 99%, Sigma-Aldrich), acetaldehyde-DNPH ($\geq$ 99%, Sigma-Aldrich), acetone-DNPH ($\geq$ 99%, Sigma-Aldrich), isobutyraldehyde ($\geq$ 99%, Sigma-Aldrich), formaldehyde-DNPH ($\geq$ 99%, Sigma-Aldrich), hydroxy acetone (95 %, Alfa Aesar), acetic acid ($\geq$99.7 %, Sigma-Aldrich), formic acid ($\geq$ 96 %, Sigma-Aldrich), (34–37 %, Trace Metal Grade, Fisher Chemical), Nitrite Standar for IC (1000 $\pm$ 4 mg/L, Sigma-Aldrich), Nitrate Standard for IC (1000 $\pm$ 4 mg/L, Sigma-Aldrich), NaOH (46–51 %, Analytical reagent grade, Fisher Chemicals), $H_2SO_4$ (95-98%, Merck). Acetonitrile (Fisher Optima), and isopropanol (Honeywell) were LC/MS grade and used as supplied. Acetone (Carlo Erba Reagents), dichloromethane (Fisher Chemical), and ether (Fisher Chemical) were HPLC grade. Tap water was purified with a Millipore MiliQ system (18.2 M$\Omega$ cm and TOC < 2 ppb). Gases were used as supplied: synthetic air (Linde, >99.999 stated purity), Helium 5.0 (Linde), and $\cdot NO_2$ (2ppm in He 5.0, Linde).

Non-commercial organic nitrates, i.e., $\alpha$-nitrooxyacetone and 1-nitrooxy-2-propanol, were synthesized and purified. $\alpha$-Nitrooxyacetone was synthesized by the nucleophilic substitution reaction of iodoacetone which was synthesized previously from chloroacetone. The ketone group from $\alpha$-nitrooxyacetone was reduced to produce 1-nitrooxy-2- propanol (see details in González-Sánchez et al., 2023).

## 3 Results

The results of aqueous phase photolysis of organic nitrates are presented stepwise. Since $NO_x$ are the known major primary products formed in the gas-phase photolysis of $RONO_2$, this process is first examined in Section 3.1 which describes the attempt to measure any formation and partitioning of $NO_x$ to the headspace of the reactor. Sections 3.2, 3.3, and 3.4 present the identified reaction products in the aqueous phase including $HNO_2$, $HNO_3$, carbonyls, organic acids, and oxidized $RONO_2$, and their associated yields. All results are reported in Table S1. Finally, Section 4 provides a detailed discussion of the mechanisms involved focusing on the fate of the nitrate group.

### 3.1 Absence of $NO_x$ in the reactor's headspace

Experiment 1 investigated isopropyl nitrate (1 mM) photolysis by analyzing the reactor's gas-phase headspace with a $NO_x$ analyzer (Fig. 1a). Prior to turning on the lamp, $\cdot NO_2$ signal increased up to $\sim$ 150 ppb, corresponding to a fraction of gas-phase isopropyl nitrate that was photolyzed inside the $NO_x$ analyzer photolytic converter (see Section S2 for further details). Once the lamp was turned on (shown in shaded blue in Fig. 1a), the aqueous-phase photolysis of isopropyl nitrate started, but no $\cdot NO$ signal was detected, while the $\cdot NO_2$ signal peaked at 800 ppb within $\sim$10 min of photolysis. However, this signal did not correspond to $\cdot NO_2$, as demonstrated by the control experiment where $\sim$ 800 ppb of $\cdot NO_{2(g)}$ were bubbled through the same volume of ultrapure water. When the lamp was turned on (shown in shaded blue in Fig. 1b), $\cdot NO_{2(g)}$ was effectively photolyzed, directly forming $\cdot NO_{(g)}$. In this experiment, barely any $\cdot NO_{2(g)}$ partitioned to the aqueous phase (confirmed by the absence of aqueous-phase $HNO_2$ or $HNO_3$, measured by HPIC), and thus the photolysis of $\cdot NO_{2(g)}$ exclusively occurred in the reactor's headspace. From this control experiment, it was concluded that if the measured $\cdot NO_2$ signal represented actual $\cdot NO_{2(g)}$

directly formed in Experiment 1, it would be photolyzed in the headspace of the photoreactor to produce measurable amounts

of $\cdot NO_{(g)}$.

Since no $\cdot NO_{(g)}$ was observed when the lamp was turned on in Experiment 1 (Fig. 1a), one can conclude that no substantial amounts of $\cdot NO_{2(g)}$ were present in the system. The signal detected as $\cdot NO_{2(g)}$ likely represented an interfering reagent. HONO cannot be this interfering reagent since the concentrations of the interference in the reactor decreased as the reaction progressed, while the measured HONO in the aqueous phase continuously increased. It likely corresponded to another volatile N-

240 containing compound that was detected by the $NO_x$ analyzer as $\cdot NO_2$ signal (as isopropyl nitrate does). Its signal could be higher than that observed for isopropyl nitrate if the compound presented less water solubility and/or if it decomposed more efficiently in the photocatalytic converter of the $NO_x$ analyzer. It is worth noting that the estimated interference for isopropyl nitrate is very low, it would represent 0.01% if equilibrium was reached.

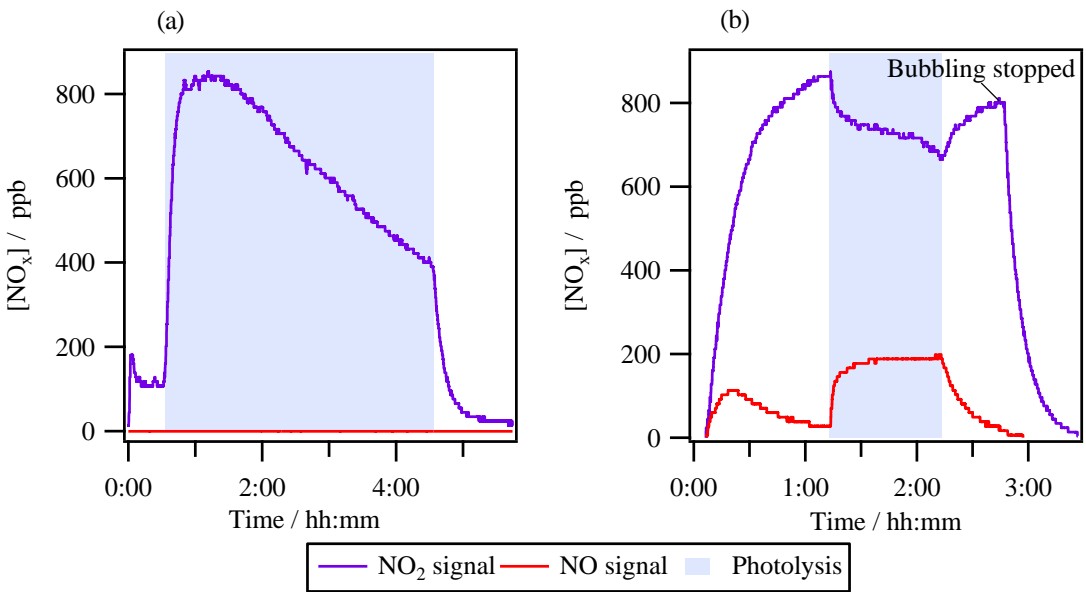

**Figure 1: Headspace time profiles of $\cdot NO_{(g)}$ and $\cdot NO_{2(g)}$ signals during a) aqueous-phase photolysis of isopropyl nitrate with $[RONO_2]_0 = 10^{-3}$ M (Exp. 1 in Table 1); and b) photolysis of $\cdot NO_{2(g)}$ bubbled in water.**

### 3.2 Formation of $HNO_2$ and $HNO_3$

$HNO_2$ and $HNO_3$ were formed during $RONO_2$ aqueous-phase photolysis. Both compounds were detected as $NO_2^-$, and $NO_3^-$ using HPIC-CD but their formation as acids was inferred by the observed fast decrease of pH (Fig. S5) and was confirmed by

250 theoretical calculations (see Section 4.1.1).

Figure 2 shows an example of $HNO_2$ and $HNO_3$ time profiles during the photolysis experiments of isopropyl nitrate (Fig. 2a), isobutyl nitrate (Fig. 2b), α-nitrooxyacetone (Fig. 2c), and 1-nitrooxy-2-propanol (Fig. 2d). $HNO_2$ could not be quantified during the aqueous-phase photolysis of α-nitrooxyacetone due to its fast hydrolysis in the HPIC system that used high pH eluents, where the molecule decomposes into lactate and $NO_2^-$ (Brun et al., 2023).

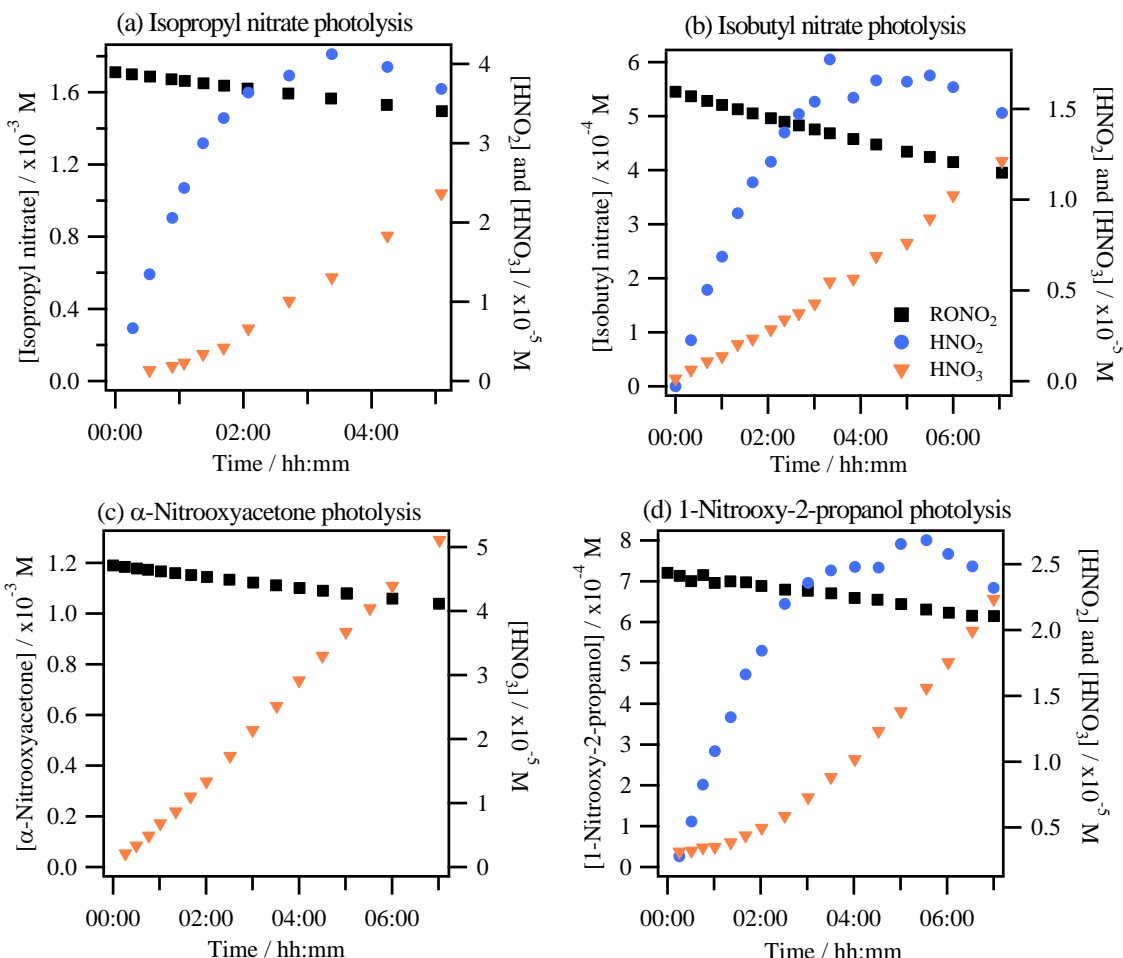

**Figure 2: RONO₂ photolysis experiments: time profiles of RONO₂, HNO₂, and HNO₃ for (a) isopropyl nitrate (Exp. 4), (b) isobutyl nitrate (Exp. 8), (c) α-nitrooxyacetone (Exp. 10), and (d) 1-nitrooxy-2-propanol (Exp. 11). HNO₂, and HNO₃ were detected as NO₂⁻, and NO₃⁻ in the HPIC-CD.**

The figure shows that $HNO_2$ was efficiently formed as a primary product during all $RONO_2$ aqueous-phase photolysis

reactions. $HNO_2$ formation slowed down over time due to its fast oxidation to $HNO_3$ whose time profiles present exponential growth due to its secondary formation. Since this conversion is fast, $HNO_3$ formation of the first aliquots has been included in the $HNO_2$ primary yields, assuming that all $HNO_3$ was formed via $HNO_2$ oxidation. The detailed chemistry of $HNO_2$/$HNO_3$ that validates this approach is discussed in Section 4.1.2. The $HNO_2$ yields ranged from 40 to 59 % for isopropyl nitrate (Exp 2 and 4), 59 to 62 % for 1-nitrooxy-2-propanol (Exp. 11 and 12), was of $31 \pm 7$ % for isobutyl nitrate (Exp. 8) and was higher

than 28 % for α-nitrooxyacetone (Exp. 10).

## 3.3 Formation of carbonyl compounds and organic acids

The formation of primary and secondary carbonyl compounds and organic acids was observed during the aqueous-phase photolysis of RONO$_2$ (Fig. 3).

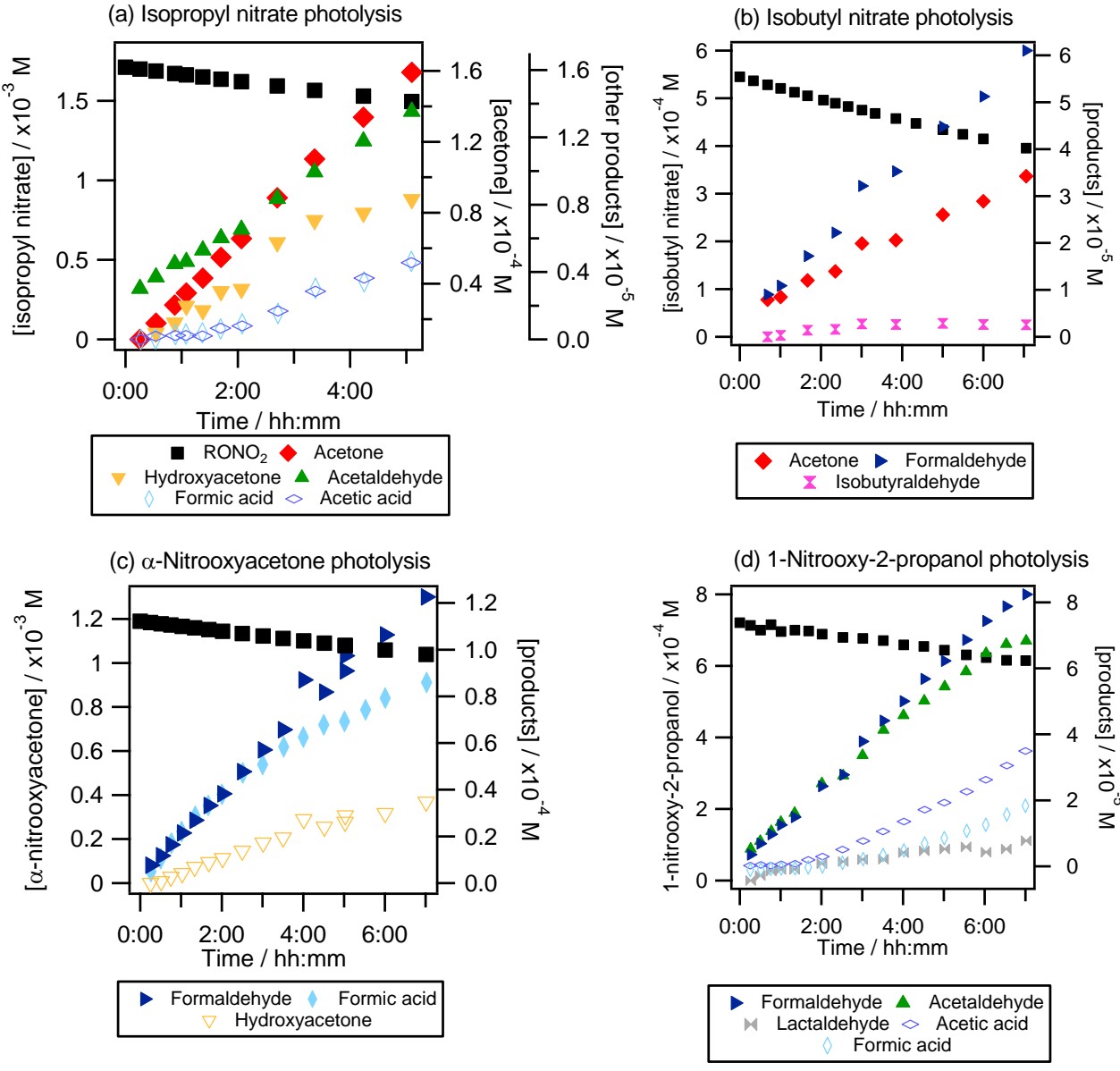

Figure 3: RONO$_2$ photolysis experiments: time profiles of RONO$_2$, carbonyl compounds and organic acids for (a) isopropyl nitrate (Exp. 4), (b) isobutyl nitrate (Exp. 8), (c) α-nitrooxyacetone (Exp. 10), and (d) 1-nitrooxy-2-propanol (Exp. 11). Plain markers are used for primary products.

For isopropyl nitrate (Fig. 3a), the main primary reaction product was acetone with yields ranging from 32 to 88 % (Exp. 2–4) and acetaldehyde was formed primarily with lower yields (5 %). Additionally, hydroxyacetone, formic acid and acetic acid were formed as secondary products. These compounds were likely formed via acetone photooxidation (Poulain et al., 2010). For isobutyl nitrate (Fig. 3b), formaldehyde and acetone were the main non-nitrogen-containing photolysis products (primary yields of 37 – 39 % and 20 – 32 %, respectively, Exp. 8–9). Additionally, isobutyraldehyde was detected as a minor product (with a primary yield of 4 – 5 %). For α-nitrooxyacetone (Fig. 3c), formaldehyde and formic acid appeared as primary products while hydroxyacetone and acetaldehyde were likely secondary products. The formation yields were found to be 96 ± 5 % and 79 ± 3 % for formic acid and formaldehyde, respectively. Other reaction products such as acetic acid and methylglyoxal were identified but not quantified due to interferences in the analyzers, caused by hydrolysis of α-nitrooxyacetone or oligomerization of methylglyoxal (see Section S4). For 1-nitrooxy-2-propanol (Fig. 3d), formaldehyde and acetaldehyde were identified as the main primary reaction products with yields of 63 – 71 % and 50 – 70 %, respectively. Furthermore, lactaldehyde was detected as a primary product with a minor yield of 8 – 14 %. Formic acid and acetic acid were observed as secondary products, likely formed via the photooxidation of formaldehyde and acetaldehyde.

## 3.4 Secondary formation of oxidized RONO$_2$

GC-MS analyses at the end of each reaction were performed to seek for nitrogen-containing organic products. For isopropyl nitrate, Fig. 4a compares the gas chromatograms obtained for the sample analyzed after 7 h of photolysis with one obtained during a control experiment of isopropyl nitrate in the dark. In both chromatograms, m/z = 46 (which corresponds to NO$_2^+$ fragment) was extracted to display chromatographic peaks related to RONO$_2$ compounds. The figure shows the formation of at least 5 oxidized RONO$_2$ molecules (IP1, IP2, IP3, IP4, and IP5), with IP3 presenting an intensity of one magnitude higher than the others.

The observed compounds were less volatile than isopropyl nitrate (which rt = 6 min, not shown in Fig. 4a) given their higher retention times and thus were probably oxidized species. The mass spectra of IP1 to IP5 confirm that all compounds were RONO$_2$ with similar chemical structures as isopropyl nitrate (included in Fig. 4b bottom right for comparison). Apart from the NO$_2^+$ fragment, other fragments observed for isopropyl nitrate were detected. Fragments such as C$_3$H$_7^+$ (m/z = 43), and C$_2$H$_4$ONO$_2^+$ (m/z = 90) were observed in IP2, IP3, and IP5 (and also IP1 for m/z = 43). Note that m/z = 43 can also correspond to an oxygenated fragment (C$_2$H$_3$O$^+$) but the resolution of 1 amu did not allow for separation from C$_3$H$_7^+$ fragments. Additionally, a specific fragment of a RONO$_2$ bearing its nitrate group on a primary carbon atom (CH$_2$ONO$_2^+$ at m/z = 76) was observed for IP1, IP3, IP4, and IP5. Since IP3 and IP5 combine this fragment with a fragment specific for the secondary nitrate group (C$_2$H$_4$ONO$_2^+$ at m/z = 90), these compounds might be dinitrates. This is the case for the most intense chromatographic peak (IP3). IP3 was thus assigned to the 1,2-propyl dinitrate molecule due to its mass spectra. Additionally, IP2 was assigned to 2-nitrooxy-1-propanol due to the C$_3$H$_7$O$^+$ and C$_2$H$_5$ONO$_2^+$ fragments (m/z = 59 and m/z = 90, respectively). These identifications are consistent with the proposed mechanism (see Section 4.2). However, the absence of standards prevented from precise identification and quantification of these compounds.

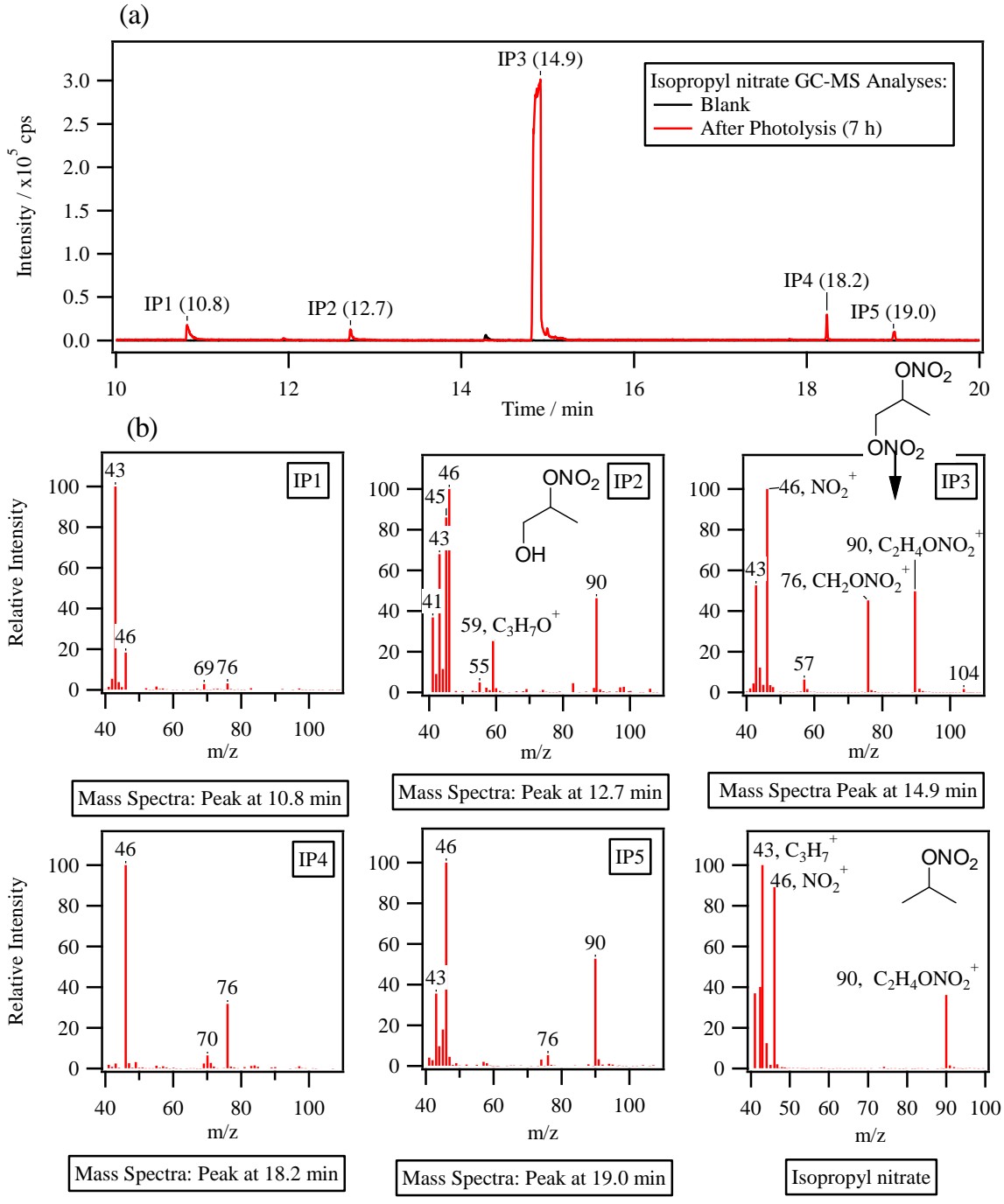

**Figure 4: Isopropyl nitrate photolysis: (a) gas chromatogram (extracted for m/z = 46, NO₂⁺) of the "end of reaction" of Exp 2 in Table 1 (after 7 hours of photolysis) and blank (isopropyl nitrate in water); (b) Mass spectra of the detected peaks.**

Hints of the formation of an oxidized $RONO_2$ were also observed in the non-derivatized UHPLC-UV analyses. An unidentified peak was detected at a retention time close to isopropyl nitrate (2.7 vs. 2.4 min). The peak presented similar UV absorption spectra to the $RONO_2$ standards (Figure S6) and was thus assigned to be IP3 (1,2-propyl dinitrate) due to its major concentrations). The compound was a secondary product since its occurrence started after 2 hours of reaction. A rough

estimation of its concentration was performed using average calibration curve parameters obtained for isopropyl nitrate, isobutyl nitrate, α-nitrooxyacetone, and 1-nitrooxy-2-propanol. Assuming that IP3 was a dinitrate, it represented 9 % of the consumed nitrogen at the end of the reaction.

The secondary formation of oxidized $RONO_2$ was also confirmed for isobutyl nitrate and 1-nitrooxy-2-propanol. For isobutyl nitrate, two unidentified peaks assigned to oxidized $RONO_2$ (IB1 and IB2) were observed by UHPLC-UV. Both compounds

present UV-Vis absorption spectra identical to isobutyl nitrate at lower retention times (1.6 min for IB1 and 3.1 min for IB2 vs 3.4 min for isobutyl nitrate) related to a higher polarity of the molecules. Their time profiles show that both compounds were formed through secondary reactions (Fig. S7). GC-MS analyses (performed after preconcentration of the sample) allowed for the detection of up to 9 oxidized $RONO_2$. For 1-nitrooxy-2-propanol, four oxidized $RONO_2$, including α-nitrooxyacetone, were observed by GC-MS. The chromatograms and mass spectra as well as comments on the identification of the formed

molecules are presented in SI (Section S5).

For α-nitrooxyacetone, no oxidized $RONO_2$ were found neither in UHPLC-UV analyses nor in GC-MS analyses.

## 4 Discussion

### 4.1 N budget during aqueous-phase $RONO_2$ photolysis

Gas-phase photolysis of $RONO_2$ is known to induce homolytic rupture of the RO–$NO_2$ bond releasing $\cdot NO_2$ to the atmosphere

with yields close to 100 % (Talukdar et al., 1997; Carbajo and Orr-Ewing, 2010). This reactivity turns $RONO_2$ into $NO_x$ reservoirs and shifts pollution transportation from the local to the regional scale. Our results show that, in the aqueous phase, a primarily formation of $HNO_2$ (with yields ranging from 31 to 62 %) is followed by a secondary formation of $HNO_3$. Therefore, one of the main questions about the aqueous-phase photolysis of $RONO_2$ is if they can (or not) regenerate $NO_x$ that would partition to the gas phase.

To address this question, we explored the viability of two different chemical pathways that lead to $NO_2^-$/$HNO_2$ and $NO_3^-$/$HNO_3$ in the aqueous phase. The first explored pathway was the direct formation of $\cdot NO_{2,(aq)}$ followed by its known aqueous reactivity (i.e., hydrolysis and reactivity towards other radicals). This pathway was rejected since $\cdot NO$ and $\cdot NO_2$ should be observable in the system under this scenario (see details in Section S6). The second explored pathway was the direct formation of $HNO_2$ in the aqueous phase. This pathway was confirmed by theoretical calculations for isopropyl nitrate aqueous-phase

photolysis. Herein, the discussion focuses on this pathway and the secondary chemistry of the photolysis products in our

system. Finally, a conclusion is given with proposed mechanisms of aqueous phase photolysis reactions of isopropyl nitrate, isobutyl nitrate, 1-nitrooxy-2-propanol, and α-nitrooxyacetone, including a detailed discussion for isopropyl nitrate.

### 4.1.1 Direct formation of HNO$_2$ in the aqueous phase

Theoretical calculations were performed to evaluate if the direct formation of HNO$_2$ is possible in the aqueous phase. The
static calculations showed that the formation of HNO$_2$ is thermodynamically favorable. Figure 5 represents the potential energy surfaces of a potential reaction pathway of isopropyl nitrate aqueous-phase photolysis, showing that it is indeed a possible reaction. Upon photon absorption, isopropyl nitrate is in the first excited state and can relax rapidly to the minimum of this state. From the excited state, it undergoes a non-radiative internal conversion back to the ground state through a degenerated point between the excited and the ground state, a conical intersection. The conical intersection has a sloped topology. The –
ONO$_2$ presents a pyramidal structure (instead of triangular) in the conical intersection. After reaching the conical intersection, the isopropyl nitrate accumulates all the photon energy as excess of vibrational energy. Indeed, this energy is in principle sufficient to cross the large barrier to form HNO$_2$ as a product. The transition state is a concerted RO—NO$_2$ dissociation and a proton transfer, with an imaginary frequency of 1241.3 cm$^{-1}$, which leads directly to the formation of acetone and HNO$_2$ as final products.

To confirm this pathway, we have performed excited state non-adiabatic dynamics in an atomistic model of isopropyl nitrate in water using QM/MM methodology. The dynamics in such model allows not only to determine the reaction hypothesis described in Fig. 5, but also to estimate the timescales at which this reaction can happen. In Fig. 6, an exemple of reactive trajectory in the excited state is depicted. Initially, the R-ONO$_2$ is in a trigonal planar conformation. Once the photon is absorbed, the group displays a pyramidal conformation that allows a non-radiative conversion from the excited to the ground state via a conical intersection. This leads directly to the dissociation of ·NO$_2$, which diffuses towards water. This is at variance with the found transition state which corresponded to a concerted ·NO$_2$ and proton transfer. Therefore, we can rather claim that the reaction happens in two steps. The interaction of ·NO$_2$ with water favors the 180-degree twist, in which nitrogen is pointing towards water molecules, favoring thus a conformation in which a proton transfer is favored, occurring in less than 1 ps. Despite this happening to be the main reaction channel, other reactions are possible in which direct formation of acetaldehyde or dissociation of HNO$_2$ in ·OH and ·NO are observed. This is due to the excess of vibrational energy of the photoproducts encapsulated in a water cavity of a diameter around 7 Å, which prevents their diffusion. Still, in longer timescales the photoproducts will either react with water or dissipate the energy to the solvent.

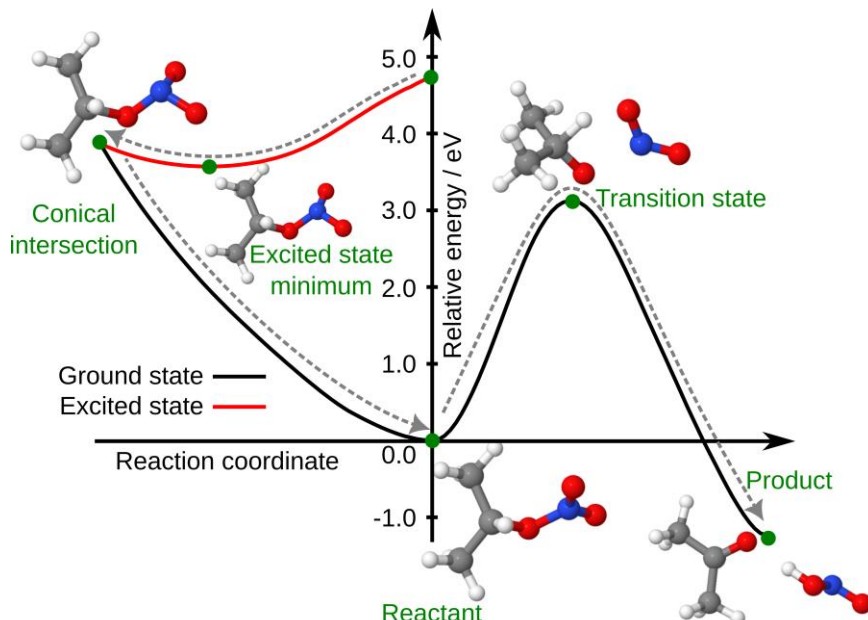

**Figure 5: Relative energy potential energy surface of isopropyl nitrate photolysis. The structures of the key points of the reaction, marked with green dots, are also depicted in the figure. The grey arrow shows the hypothetical reaction: (1) the molecule is in an excited state after photon absorption, traveling first to the minimum energy structure of the excited state (2) and accessing a conical intersection (3) through a barrierless surface. The molecule is then back to the reactant (4) with excess of vibrational enegy, that allows to cross a transition state (5) up to the products (6). The reactant's energy has been arbitrarily placed at 0.0 eV. Calculations were performed using a DFT/TDDFT model in implicit solvent.**

The gas-phase photolysis of isopropyl nitrate was also studied using the same type of dynamics. In the gas phase, the photochemical pathway through a conical intersection until the formation of $\cdot NO_2$ radical happens in a similar manner to the aqueous solution. In the gas phase, however, the absence of water cavity prevents the resulting fragments from reorienting and reacting. Indeed, similar to an explosion, the two resulting fragments $\cdot NO_2$ and $RO\cdot$ diffuse in opposite directions before any proton transfer can occur between them. This explains the observed direct formation of $\cdot NO_{2,(g)}$ (Talukdar et al., 1997). In contrast, in the cavity, collisions with the solvent are frequent, and thus, the photolysis likely follows the pathway with the minimum energy barrier, that leads to the formation of $HNO_2$ and acetone.

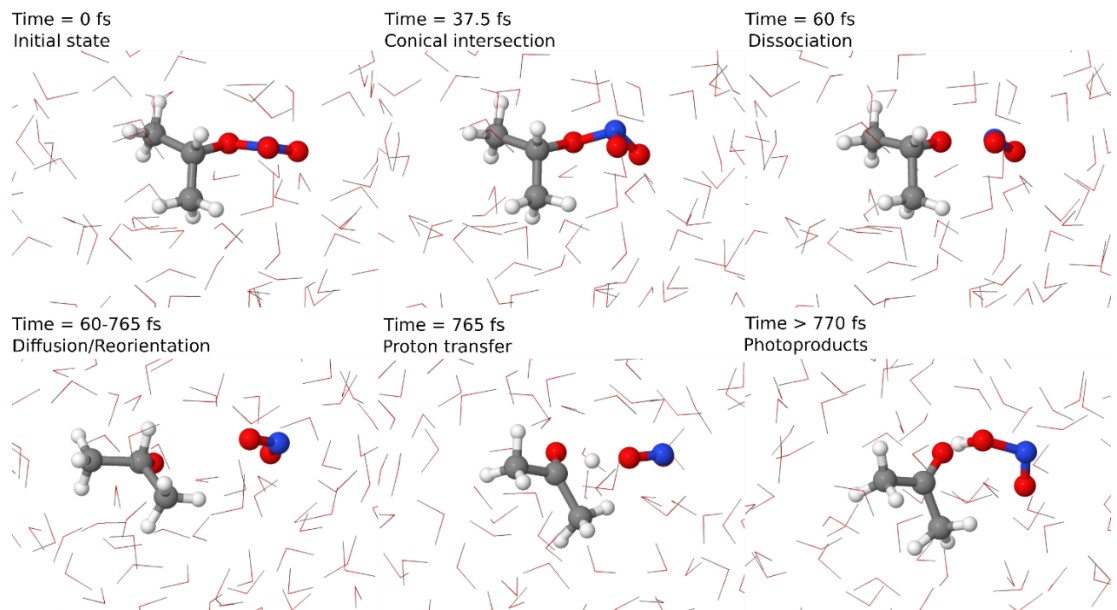

**Figure 6: Representative reactive trajectories for the formation of acetone and HNO₂ with the main reaction steps depicted. Timescales are just indicative.**

These calculations agree with our observations where the photolysis of isopropyl nitrate provided the direct concomitant formation of HNO₂ and acetone as shown in Fig. 7. Furthermore, minor trajectories where the carbon backbone structure of isopropyl nitrate breaks leading to the formation of acetaldehyde and other species have been also experimentally observed since acetaldehyde was determined to be a primary product with low yields (~4 %).

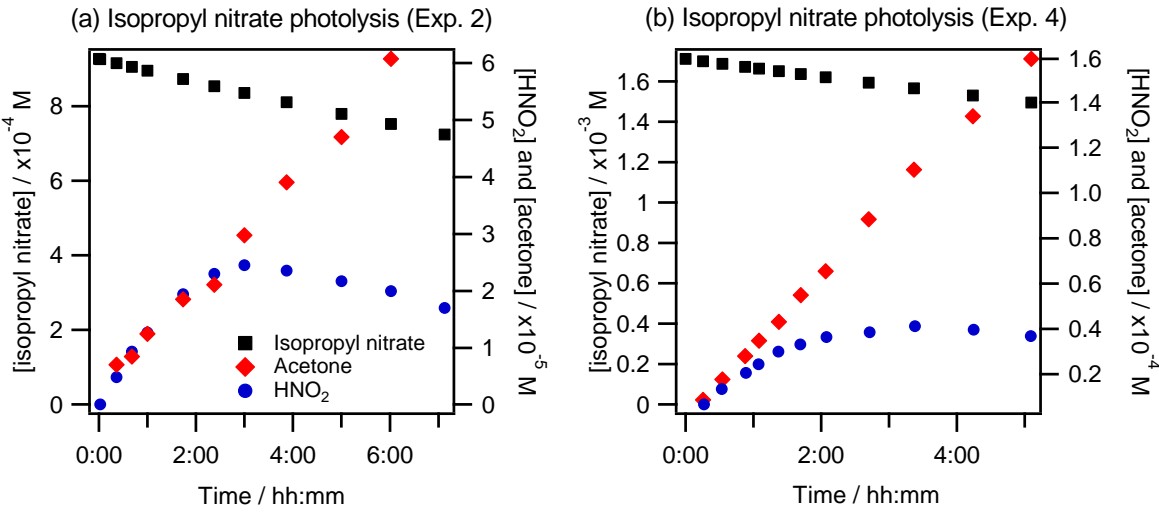

**Figure 7: Concomitant acetone and HNO₂ formation during isopropyl nitrate photolysis at two different initial concentrations (a) 0.57 mM (Exp.2) and (b) 1.42 mM (Exp. 4).**

### 4.1.2 Secondary chemistry of HNO₂ in the aqueous phase

Once formed in the solution, $HNO_2$ was highly reactive as shown by its time profiles (in Fig. 7). It may disproportionate to

yield $\cdot NO$ and $\cdot NO_2$ (R1).

$$2HNO_2 \rightarrow \cdot NO + \cdot NO_2 + H_2O \tag{R1}$$

However, this reaction is quite slow under our experimental conditions (rate constant of 28.6 M$^{-1}$ s$^{-1}$, Vione et al., 2004).
Nevertheless, considering the lamp actinic flux, the photolysis/photooxidation of $HNO_2$ was likely its major sink. The
photolysis of $HNO_2$ and $NO_2^-$ is known to form $\cdot NO$ and a $\cdot OH$ radical (R2–4) (Mack and Bolton, 1999; Fischer and Warneck,

1996; Kim et al., 2014). $HNO_2$ can also decompose due to the additional energy of the $RONO_2$ photolysis.

$$NO_2^- + h\upsilon \rightarrow \cdot NO + \cdot O^- \tag{R2}$$

$$HNO_2 + h\upsilon \rightarrow \cdot NO + \cdot OH \tag{R3}$$

$$\cdot O^- + H_3O^+ \leftrightarrow \cdot OH + H_2O \tag{R4}$$

Additionally, $\cdot OH$ radicals readily react with $NO_2^-$ and with $HNO_2$ (rate constants of $1.0 \cdot 10^{10}$ and $2.6 \cdot 10^9$ M$^{-1}$ s$^{-1}$, respectively)

(Mack and Bolton, 1999; Kim et al., 2014).

$$NO_2^- + \cdot OH \rightarrow \cdot NO_2 + OH^- \tag{R5}$$

$$HNO_2 + \cdot OH \rightarrow \cdot NO_2 + H_2O \tag{R6}$$

$HNO_3$ can then be formed through $\cdot NO_2$ hydrolysis (R7) (Finlayson-Pitts and Pitts Jr., 2000).

$$2 \cdot NO_2 + H_2O \rightarrow HNO_2 + HNO_3 \tag{R7}$$

Another pathway can be initiated by the reaction of $\cdot NO$ with $HO_2\cdot$ radicals to yield peroxynitrite (R8). The latter can isomerize
into $HNO_3/NO_3^-$ (R9) or decompose yielding $\cdot NO_2$ and $\cdot OH$ radicals (R10).

$$\cdot NO + HO_2 \cdot \rightarrow HOONO \tag{R8}$$

$$HOONO \rightarrow NO_3^- + H^+ \tag{R9}$$

$$HOONO \rightarrow \cdot NO_2 + \cdot OH \tag{R10}$$

$HO_2\cdot$ radicals were likely formed by the photooxidation of organic compounds. Since $\cdot OH$ radicals were formed through
$HNO_2/NO_2^-$ photolysis they could attack the organic molecules present in the photoreactor (i.e., the $RONO_2$, as no scavenger
was used). Upon oxygen addition, the $\cdot OH$ attack yielded peroxy radicals. The formation of peroxy radicals was confirmed by
the dissolved oxygen time profiles: during each photolysis experiment, dissolved $[O_2]$ underwent slight decay due to the
reaction of alkyl radicals $(R\cdot)$ and oxygen (Fig. S8).

Peroxy radicals can readily react with $\cdot NO$ to form peroxynitrites (ROONO) that can isomerize to $RONO_2$ (R11). Additionally,
peroxy radicals react with $\cdot NO_2$ forming peroxynitrates (ROONO₂) (Goldstein et al., 2004).

$$ROO \cdot + \cdot NO \rightarrow ROONO \rightarrow RONO_2 \tag{R11}$$

$$ROO \cdot + \cdot NO_2 \rightarrow ROONO_2 \tag{R12}$$

The rate constants observed for reactions R11 and R12 range, respectively, from $2.8 \cdot 10^9$ to $3.5 \cdot 10^9$ M$^{-1}$ s$^{-1}$ (with R being

$(CH_3)_2CCH_2-$ and $CH_3-$), and from 0.7 to $1.5 \cdot 10^9$ M$^{-1}$ s$^{-1}$ (with R = $(CH_3)_2CCH_2-$, $CH_3-$, and c-$C_5H_9-$). Therefore, these

reactions could readily occur under our experimental conditions, but not in the first step as they would be limited by the formation of ·OH radicals to be initiated.

In our experiments, the formation of oxidized RONO$_2$ during isopropyl nitrate and isobutyl nitrate photolysis was confirmed by GC-MS, and UHPLC-UV analyses (Fig. 4 and Section S4). The possibility to form oxidized RONO$_2$ via the aforementioned

reactions is consistent with the substantial number of compounds displaying the NO$_2^+$ fragment found by GC-MS analyses (up to 6 compounds for isopropyl nitrate photolysis and up to 8 for isobutyl nitrate). Nevertheless, ROONO$_2$, if formed, were not detected due to their thermolysis during the analysis.

During isopropyl nitrate photolysis, the main formed oxidized RONO$_2$ (IP3 in Fig. 4) was suspected to be a dinitrate (1,2-propyldinitrate) since its mass spectra conjugate mass fragments that correspond to both primary (m/z = 76, CH$_2$ONO$_2^+$) and

secondary nitrate groups (m/z = 90, CH(ONO$_2$)CH$_3^+$). The formation of this compound through secondary photochemistry of HNO$_2$/NO$_2^-$ agrees well with the observed secondary time profile of this product. An equivalent compound was observed during isobutyl nitrate photolysis (IB6 in Fig. S4.1).

## 4.2 Proposed chemical mechanisms

### 4.2.1 Isopropyl nitrate aqueous-phase photolysis proposed mechanism

Combining all the reactions mentioned in the discussion, Fig. 8 proposes a complete mechanism of isopropyl nitrate aqueous-phase photolysis.

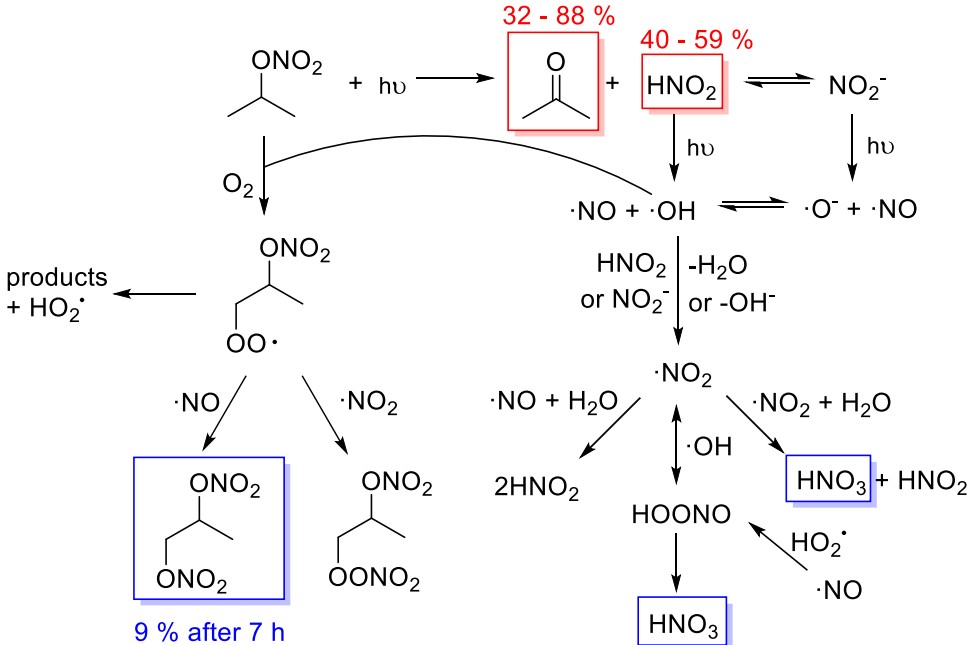

**Figure 8: Proposed mechanism of isopropyl nitrate photolysis in the aqueous phase. In red are the measured primary products with their molar yields. In blue are the measured secondary products.**

Isopropyl nitrate photolyzes into acetone and nitrous acid. Nitrous acid undergoes equilibrium in the aqueous phase with nitrite. Both $HNO_2$ and $NO_2^-$ can undergo photolysis yielding $\cdot NO$ and $\cdot OH$ radicals (R5 to 7). $\cdot OH$ radicals can react with isopropyl nitrate yielding an alkyl radical that upon oxygen addition forms a peroxy radical. The peroxy radical can decompose into products (i.e., acetone, formic acid, acetic acid, hydroxy acetone, and acetaldehyde which also could be issued from acetone photooxidation), or react with $\cdot NO$ or $\cdot NO_2$, to form a dinitrate or a peroxynitrate. The dinitrate likely corresponds to the

compound detected by GC-MS (IP3 in Fig. 4) and is formed secondarily in agreement with the proposed mechanism. Additionally, IP3 was estimated to account for 18 % of the reactive N at the end of the reaction, in agreement with the 20 % of isopropyl nitrate estimated to undergo $\cdot OH$ oxidation. Furthermore, $HNO_3$ is formed through secondary reactions such as $\cdot NO_2$ hydrolysis (R1) or peroxynitrite isomerization (R11).

### 4.2.2 Isobutyl nitrate and 1-nitrooxy-2-propanol aqueous-phase photolysis proposed mechanism

The primary formation of $HNO_2$ was also observed during the photolysis of isobutyl nitrate and 1-nitrooxy-2-propanol in the aqueous phase (Fig. 2). The determined yields were 31 % and 59–62 %, for isobutyl nitrate (Fig. 9a), and 1-nitrooxy-2-propanol (Fig. 9b), respectively. Although no DFT calculations were performed specifically for these molecules, they likely undergo a similar photolysis process to the one detailed for isopropyl nitrate, where an adjacent hydrogen atom is captured by the $-NO_2$ leaving moiety (as shown in Fig. 6).

Nevertheless, the formation of carbonyl products concomitant to $HNO_2$ was different from those expected from the main isopropyl nitrate mechanism. The corresponding carbonyl compounds were only observed in minor proportions: yields of 5 % isobutyraldehyde and 8–10 % lactaldehyde were obtained respectively for isobutyl nitrate and 1-nitrooxy-2-propanol. The major carbonyl products were formed after the breakdown of the organic chain, probably due to the excess energy the molecules have after light absorption. This pathway has been observed during the isopropyl nitrate calculations although as a

minor pathway, leading to the formation of acetaldehyde. Figure S9 clearly shows that the carbonyl products formed concomitantly to $HNO_2$ were acetone and formaldehyde (yields of 20–32 % and 37–39 %, respectively) during isobutyl nitrate photolysis, and formaldehyde and acetaldehyde (yields of 63–71 % and 50–70%, respectively) during 1-nitrooxy-2-propanol photolysis.

   The proposed pathways for their photolysis are given in Fig. 9. Further studies should be conducted to understand the

breakdown of the organic chain.

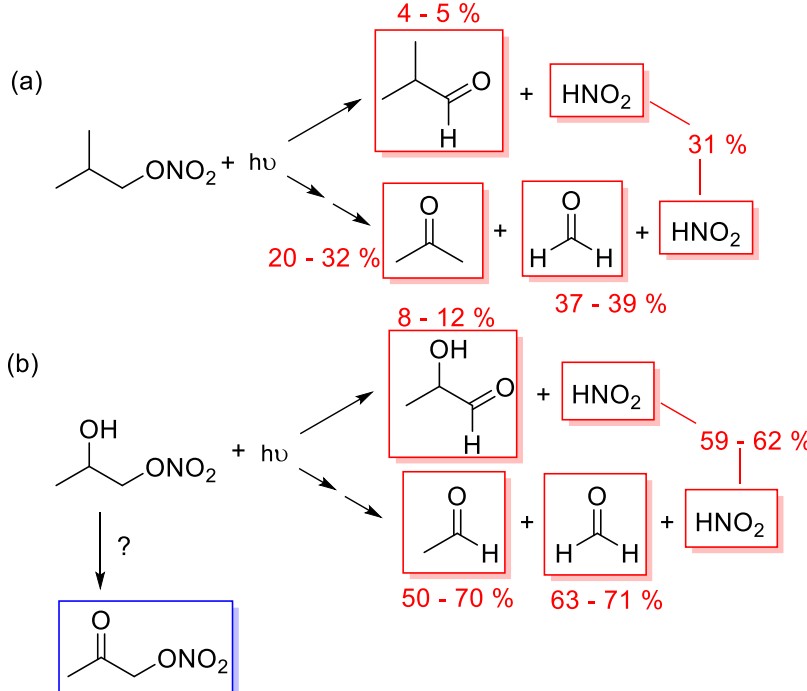

**Figure 9: Proposed mechanisms of aqueous-phase photolysis of (a) isobutyl nitrate, and (b) 1-nitrooxy-2-propanol. In red are the measured primary products, and in blue, the detected secondary products.**

During isobutyl nitrate photolysis, two oxidized $RONO_2$ were observed, these compounds were likely formed through the $\cdot OH$ oxidation of isobutyl nitrate initiated by $HNO_2/NO_2^-$ aqueous-phase photochemistry (as it occurs in isopropyl nitrate photolysis). During 1-nitrooxy-2-propanol photolysis, the secondary formation of α-nitrooxyacetone was observed (Fig. 9b).

### 4.2.3 α-Nitrooxyacetone aqueous-phase photolysis proposed mechanism

During α-nitrooxyacetone photolysis, $NO_2^-$ could not be measured due to its base-catalyzed hydrolysis in the HPIC system (at pH = 12, Brun et al., 2023), but $NO_3^-$ was quantified and showed a secondary formation. Primary formation of $HNO_2$ was therefore expected with a minimum yield of 28 %. The complete mechanism could not be elucidated for this molecule, since some of its primary products could not be quantified (i.e., methylglyoxal, and acetic acid). Nevertheless, the mechanism appears to be similar to those of isobutyl nitrate and 1-nitrooxy-2-propanol. Identification of methylglyoxal indicates one pathway leading to its direct formation along with $HNO_2$ (homologous mechanism as in Fig. 5). However, the detection of significant quantities of formic acid, formaldehyde, and identification of acetic acid suggests that the molecule tends to break at different points (similar to isobutyl nitrate and 1-nitrooxy-2-propanol). Further analytical efforts or modeling should be conducted to clearly identify how the molecule undergoes rupture by photolysis.

# 5 Conclusions and atmospheric implications

This work has investigated the fate of the nitrate group during the aqueous-phase photolysis of four $RONO_2$ species: isopropyl nitrate, isobutyl nitrate, 1-nitrooxy-2-propanol, and α-nitrooxyacetone. Our findings suggest a completely different reactivity from the gas phase one. While $RONO_2$ releases $NO_x$ back to the atmosphere upon photolysis in the gas phase, $HNO_2$ is directly formed in the aqueous phase.

$HNO_2$ was detected as a primary reaction product along with others such as carbonyl compounds or organic acids. The direct formation of $HNO_2$ by aqueous-phase photolysis was confirmed by DFT theoretical calculations and was supported by the absence of direct $\cdot NO_2$ formation, due to solvent cage effects.

Therefore, aqueous-phase photolysis of $RONO_2$ represents both a sink of $NO_x$ and a source of atmospheric $HNO_2$ (or HONO). The latter is an important precursor of $\cdot OH$ and $\cdot NO$ radicals. During our experiments, these secondarily formed radicals were shown to be trapped in the aqueous phase, producing $HNO_3$ and functionalized $RONO_2$. In the atmosphere, this reactivity can potentially contribute to the sink of $NO_x$, a source of $\cdot OH$ radicals in condensed phases, and an additional source of $SOA_{aq}$. Aqueous-phase photolysis has been reported to be negligible in the $RONO_2$ sinks in the atmosphere due to the hindering effect of the "solvent cage" (González-Sánchez et al., 2023). Nevertheless, the mechanisms of this reactivity might be relevant for more significant reactions such as the aqueous-phase $\cdot OH$ oxidation of $RONO_2$, or potentially their heterogeneous photolysis. Therefore, further work should be done to better assess the role of $RONO_2$ in $NO_x$ sink and transport, in the formation of atmospheric HONO and SOA.

*Data and code availability*. Data related to this article are available at https://doi.org/10.7910/DVN/USWU6V (González-Sánchez, 2023). Data related to the theoretical calculations can be requested from Miquel Huix-Rotllant (miquel.huixrotllant@univ-amu.fr).

*Author contributions*. JMGS performed all experiments and treated all experimental data. MHR developed the quantum chemistry model and conducted the calculations. JMGS, NB, and CD developed the HPIC-CD method. JM build the $NO_x$ analyzer experimental setup. JMGS and SR developed the UHPLC-UV method. JMGS and AD developed the GC-MS method. JMGS and JLC performed the organic synthesis of $RONO_2$. AM and JLC coordinated the work. JMGS, MHR and AM wrote the article with inputs from all co-authors.

*Competing interests*. The authors declare that they have no conflict of interest.

*Financial support*. This project has received funding from the European Union's Horizon 2020 research and innovation programme under the Marie Skłodowska-Curie (grant no. 713750). It has been carried out with the financial support of the Regional Council of Provence-Alpes-Côte d'Azur and with the financial support of the A*MIDEX (grant no. ANR- 11-IDEX-

0001-02), funded by the Investissements d'Avenir project funded by the French Government, managed by the French National Research Agency (ANR). This study also received funding from the French CNRS-LEFE-CHAT (Programme National-Les Enveloppes Fluides et l'Environnement-Chimie Atmosphérique – Project "MULTINITRATES") and from the program ANR-PRCI (ANR-18-CE92-0038-02) – Project "ARAMOUNT".

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
