# Peer review of "Direct formation of HONO through aqueous-phase photolysis of organic nitrates"

_EGUsphere, 2023_

## Author Response (AR1)

The authors thank the reviewers for their positive feedback on our work and their helpful comments and questions. We have addressed all comments and corrections as detailed below, and we modified the manuscript accordingly.

**Referee #1**

**Major issue**

**I didn't realize until the very end of the manuscript (Line 458) that the previous paper by the authors indicated that aqueous phase photolysis is too slow to be an important atmospheric fate for these particular organonitrates. I had to read this previous paper by the authors to understand this implication. On the other hand, that previous paper also indicates that other organonitrates (more water soluble ones) could undergo significant aqueous phase photolysis. I think the present paper would be well-served by rewriting the introduction to more thoroughly explain the findings of the previous paper and then specifically indicating that the particular organonitrates under study in the present paper are serving as proxies for other organonitrates for which the mechanism might be atmospherically significant. In its current form, it's not clear that the work is directly relevant at all to actual atmospheric processes since it focusses on the specific organonitrates under study.**

The investigated $RONO_2$ are not sufficiently water-soluble (referring to their Henry's Law equilibrium) to make their aqueous-phase reactivity relevant in the atmosphere. As pointed out by the reviewer, these $RONO_2$ were used as proxies.

Atmospherically relevant $RONO_2$ can exhibit much higher water solubilities and thus, partition effectively into the atmospheric aqueous phase. Nevertheless, in our previous paper, we estimated that aqueous-phase reactivity is mostly lead by ·OH oxidation, with photolysis playing a minor role (< 1%). Additional experimental data would be necessary to confirm our estimations.

However, even if aqueous phase photolysis is a minor atmospheric sink, even for large $RONO_2$, we believe that understanding the reaction mechanisms of $RONO_2$ aqueous-phase photolysis is of high atmospheric relevance for two reasons: 1) it represents one of the missing HONO sources, and 2) it will be essential to gain a better understanding of other more relevant sinks, such as aqueous-phase ·OH oxidation, and other unexplored but potentially relevant sinks, such as heterogeneous photolysis.

The introduction was extended to include these points:

*" This work intends to address these questions for the aqueous-phase photolysis of $RONO_2$. While aqueous-phase photolysis is a minor sink when compared to ·OH oxidation (González-Sánchez et al., 2023), the investigation of the photolysis mechanisms is of higher importance as it is a first step towards the fundamental understanding of $RONO_2$ photooxidation pathways.*

*To explore the aqueous-phase photolysis of $RONO_2$, the fate of four molecules (i.e., isopropyl nitrate, isobutyl nitrate, α-nitrooxyacetone, and 1-nitrooxy-2-propanol) was experimentally investigated. These $RONO_2$ served as proxies to understand the fate of the nitrate group. The two alkyl nitrates are simple molecules, simplifying the comprehension of mechanisms related to nitrate group reactivity. Furthermore, the other two investigated $RONO_2$ are polyfunctional, they combine highly relevant functional groups (hydroxy and carbonyl groups) with the nitrate group, allowing for the assessment of their influence on the reactivity. The aqueous-phase photolysis primary and secondary reaction products were identified and quantified, and the fate of the nitrate group was elucidated with support from theoretical calculations. The atmospheric implications of these findings are discussed. "*

**Specific comments**

**Throughout: I understand why the authors use the more typical chemical formula representation for nitrous acid (HONO) in the title and abstract, but then use HNO2 in the body (because it easier to explain its relationship to other nitrogen oxides). However, I do think this will be confusing to casual readers. What about just using "nitrous acid" in the title and abstract and then the first time it is used in the body of the manuscript, define it as HNO2?**

It is true it could be confusing. We have included the definition of nitrous acid as both HONO and HNO2 in the abstract to clarify this. We would like to keep the title as "Direct formation of HONO... ".

**Lines 32-34: I find this sentence confusing. Are the authors referring to a higher proportion of RONO2 forming from NOx presently due to the gradual decreasing importance of other sinks for NOx?**

As a result of lower $NO_x$ emissions, and thus lower $NO_x$ concentrations, these species form $RONO_2$ rather than $HNO_3$. We have revised the paragraph for better clarity:

"*Furthermore, $RONO_2$ also participate in $NO_x$ removal from the atmosphere can occur either through their deposition to the Earth's surface or by transformation into a less reactive chemical compound, such as nitric acid (Hu et al., 2011; Nguyen et al., 2015). Therefore, their atmospheric reactivity and fate must be considered to accurately predict pollution transport on a regional scale. This is especially important for world regions experiencing decreasing $NO_x$ levels, such as Europe and North America, where the relative importance of $RONO_2$ in $NO_x$ transport and removal is growing due to the increase of the overall transformation of $NO_x$ into $RONO_2$ (Romer Present et al., 2020).*"

**Line 127: typo: "preconcentrated"**

Done

**Line 210: Isn't this a lot of "background" signal? Could this unknown species be undergoing other reactions that could compromise the analysis? What's an estimate for its relative concentration compared to the known organonitrate species?**

It is not possible to estimate the unknown species concentration since it depends on its water solubility and its response in the photocatalytic converter of the $NO_x$ analyzer.

**Figure 2: From the comment on Line 230, it seemed as if the authors were going to discuss, if not model, the interesting HNO2 and HNO3 time dependence in this figure in Section 3.5. However, there is no Section 3.5. In any case, it would be good to try to rationalize this data more quantitatively.**

The correct reference on that line should be Section 4.1.1, where we confirm that $HNO_2$ is directly formed. A discussion on the time dependence of both $HNO_2$ and $HNO_3$ is given in Section 4.1.2 although no modeling was performed.

**Figure 10: I don't understand the yields given in this figure. Does the 96% refer to the sum of the products in the second channel, which could theoretically have yields of 300% if it were the only product producing channel? Otherwise, it seems as if the second and third channels are yielding more than 100%...**

Thanks for pointing out this, we had not noticed this inconsistency. In the single experiment conducted for α-nitrooxyacetone, we determined a 96 ± 5 % (molar) yield of formic acid, and a 57 ± 3 % yield of formaldehyde. Indeed, it is possible that the determined yield of formic acid was overestimated. Possible explanations could include the rapid oxidation of formaldehyde into formic acid or potential interferences in the HPIC analyses. Nevertheless, it might be possible that the proposed mechanism is incorrect, it is

difficult to elucidate this mechanism while some of the primary products, such as acetic acid and methylglyoxal, could not be quantified. We have revised Section 4.23, to simply outline what is known about the α-nitrooxyacetone photolysis mechanism, and Fig. 10 has been removed.

*"During α-nitrooxyacetone photolysis, $NO_2^-$ could not be measured due to its base-catalyzed hydrolysis in the HPIC system (at pH = 12, Brun et al., 2023), but $NO_3^-$ was quantified and showed a secondary formation. Primary formation of $HNO_2$ was therefore expected with a minimum yield of 28 %. The complete mechanism could not be elucidated for this molecule, since some of its primary products could not be quantified (i.e., methylglyoxal, and acetic acid). Nevertheless, the mechanism appears to be similar to those of isobutyl nitrate and 1-nitrooxy-2-propanol. Identification of methylglyoxal indicates one pathway leading to its direct formation along with $HNO_2$ (homologous mechanism as in Fig. 5). However, the detection of significant quantities of formic acid, formaldehyde, and identification of acetic acid suggests that the molecule tends to break at different points (similar to isobutyl nitrate and 1-nitrooxy-2-propanol). Further analytical efforts or modeling should be conducted to clearly identify how the molecule undergoes rupture by photolysis."*

**Referee #2**

**Major Points.**

**The experiments in Figure 1 show the concentration of NOx evolved from water when (a) aqueous phase RONO2 are irradiated and (b) when pure water is irradiated while bubbling NO2 through solution. The results show that irradiation of RONO2 solutions produces NO2, while in the case of irradiating the system where NO2 is bubbled through the water, both NO2 and NO are measured. The authors use the latter experiment as evidence to support their hypothesis that the NO2 evolved from the RONO2 solution is not NO2, but rather due to an interference with the chemiluminescence NOx analyzer that only appears when the organic nitrate is photolyzed. However, they do not go into further detail about what this interference is. Do they imply that HONO is the interference or an organic nitrate product? I am not convinced yet that HONO is the interference since the blue light converter should have a low HONO-to-NO conversion efficiency (can you measure this to put a number on it?). Also, if it was HONO, why would the yield decrease as the reaction progresses when the pH of the solution actually decreases and would start favoring nitrite protonation. Can you predict the amount of HONO that would form based on the nitrite concentrations and Henry's law to see if that number matches the interference observed in Figure 1a? I am also not convinced that the interference is an organic nitrate product, if that is what the authors imply. The pure organic nitrate species shows a ~17% interference as shown by the signal in Figure 1a prior to photolysis. The ~900 ppb of signal formed is too high, even if you had 100% yield of a RONO2 product (assuming it would have a similar interference profile). Unless the authors show the interference is HONO, I believe one cannot discount the possibility that the signal is mostly due to NO2 in this experiment for the following rationale: Experiment in Fig 1b is not a perfect control since (1) it lacks the reactive intermediates that are present in a solution of irradiated RONO2 and (2) NO2 is very insoluble in water and it is virtually impossible to dissolve it in water from the gas phase. NO2 formed as a photoproduct during the photolysis of RONO2 starts out hydrated in the solvent cage, and therefore will be in proximity with reactive intermediates to react with (this is the point of the HONO formation mechanism proposed). It would make sense that NO2 formed as a photoproduct from RONO2 could itself undergo photolysis to form NO. However, I could imagine that NO would be highly reactive with other reactive oxygen species produced in solution. For example, perhaps the reason why NO is not observed in Figure 1a is because NO is reacting with RO2 species**

**present in solution, which would yield NO2 as a product. Also, there is some work on nitrate photolysis (Scharko et al. EST, 2014) that made a similar suggestion and used a kinetic model to show that hydrolysis of solvated NO2 formed during nitrate photolysis was responsible for at least some of the HONO formed. I note in the SI that the authors seem to rule out the NO2 hydrolysis pathway stating that the nitrate expected to be formed from this reaction would not exhibit the nitrate profile seen in Figure 2. Maybe this is correct, but the organic nitrate system is in some ways more complicated than the nitrate system with a lot more secondary chemistry, so I think one cannot really with certainty say what the HONO formation mechanism is without modeling the kinetics.**

We agree with the reviewer that the very high signal obtained in Figure 1 was surprising. However, as pointed out by the reviewer, HONO cannot be the interfering reagent since the concentrations of the interference in the reactor decreases as the reaction progresses.

While $\cdot NO_2$ is indeed formed during the initial stages of the photolysis, our theoretical calculations demonstrated that within a few fs, $\cdot NO_2$ transforms into HONO. According to the calculations, negligible $\cdot NO_2$ diffuses from the solvent cage. Some secondary $\cdot NO_2$ could escape and reach the reactor's headspace, and there, it would partially undergo photolysis into $\cdot NO$. While it is true that the experiment presents more reactive intermediates that can react with $\cdot NO$ compared to the control experiment, these intermediates are primarily concentrated in the reactor's aqueous phase. Since both the reactor's aqueous phase and headspace are illuminated, we believe that $\cdot NO$ would be detectable under such high and sudden increase of $\cdot NO_2$ concentrations. Furthermore, secondary $\cdot NO_2$ would likely exhibit a different profile than the one observed, as its concentration would follow that of HONO. Hence, it is likely that all secondary $\cdot NO_2$ remains in the aqueous phase.

Our hypothesis is that the interference is an organic nitrogenated species formed directly during the isopropyl nitrate photolysis. Its signal could be higher than that observed for isopropyl nitrate if the compound presents less water solubility and/or if it decomposes more efficiently in the photocatalytic converter. Note that the estimated interference for isopropyl nitrate is very low, it would be 0.01% if equilibrium was reached.

We have changed the text to clarify all this:

*"When the lamp was turned on (shown in shaded blue in Fig. 1b), $\cdot NO_{2(g)}$ was effectively photolyzed, directly forming $\cdot NO_{(g)}$. In this experiment, barely any $\cdot NO_{2(g)}$ partitioned to the aqueous phase (confirmed by the absence of aqueous-phase $HNO_2$ or $HNO_3$, measured by HPIC), and thus the photolysis of $\cdot NO_{2(g)}$ exclusively occurred in the reactor's headspace. From this control experiment, it was concluded that if the measured $\cdot NO_2$ signal represented actual $\cdot NO_{2(g)}$ directly formed in Experiment 1, it would be photolyzed in the headspace of the photoreactor to produce measurable amounts of $\cdot NO_{(g)}$.*

*Since no $\cdot NO_{(g)}$ was observed when the lamp was turned on in Experiment 1 (Fig. 1a), one can conclude that no substantial amounts of $\cdot NO_{2(g)}$ were present in the system. The signal detected as $\cdot NO_{2(g)}$ likely represented an interfering reagent. HONO cannot be this interfering reagent since the concentrations of the interference in the reactor decreased as the reaction progressed, while the measured HONO in the aqueous phase continuously increased. It likely corresponded to another volatile N-containing compound that was detected by the $NO_x$ analyzer as $\cdot NO_2$ signal (as isopropyl nitrate does). Its signal could be higher than that observed for isopropyl nitrate if the compound presented less water solubility and/or if it decomposed more efficiently in the photocatalytic converter of the $NO_x$ analyzer. It is worth noting that the estimated interference for isopropyl nitrate is very low, it would represent 0.01% if equilibrium was reached."*

**The last major point is regarding the theory results. I found section 4.1.1 very difficult to follow and too brief to describe the nuances of these experiments. The authors start the paragraph off by stating the conclusions first, rather than leading the reader through the results on the way to a conclusion. I recommend they start this paragraph out by proposing their hypothesis followed by how they want to test that hypothesis. Then, they can discuss their results and discuss whether the reactions are thermodynamically feasible. Discussion of the results would also benefit from a clearer graphics. For example, it may be easier to follow Figure 5 if the structures of certain transition states were included in the figure. It would be helpful if somehow the lines connecting the various energy states could more clearly represent the direction of the reaction pathway. For example, it seems as if the lines suggest that excitation of ground state RONO2 relaxes back down to the ground state, which somehow reacts via TS #4 onto the product. Is it possible to move from structure 2 to 4, because moving from structure 0 or 3 (?) to products via TS #4 has a 32 kcal/mol energy barrier, which is quite high.**

We have improved the description of section 4.1.1 and Figure 5 according to the indications of the reviewer, by adding the structures of intermediates and performing real potential energy surface calculations. Figure 5 represents the trajectory in which we found that the transition state is concerted $RO—NO_2$ dissociation and proton transfer.

We confirm partially this hypothesis by doing the quantum dynamics simulations, in which we show a film of the reaction occurring at different times. Indeed, the concerted TS is never visited, but rather the dissociation occurs in two steps. First the formation of $RO·$ and $·NO_2$ radicals, then a reorientation of the fragments in the water cavity and finally the proton transfer. This makes the overall reaction very efficient, to judge the sub-picosecond timescales at which these steps happen.

**Also, for Figure 5 were the water molecules in this calculation treated explicitly or was a force field used? I note that in the methods section, the authors mention doing gas-phase and aqueous phase calculations. However, I only see calculations done in the presence of water.**

The water molecules are treated at the MM level using the well-known force field for water called TIP3P model, as implemented in Amber99 force field. These details are now described in section 2.5. We have explicitly clarified this in the revised manuscript.

The article focuses on the reactivity in aqueous solutions; therefore we chose to show only graphs related to models of aqueous solutions. The gas phase results are used for dynamic calculations, like those shown in Fig. 5. The gas phase results are described in the following paragraph in section 4.1.1, which stem from our calculations:

*"The gas-phase photolysis of isopropyl nitrate was also studied using the same type of dynamics. In the gas phase, the photochemical pathway through a conical intersection until the formation of $·NO_2$ radical happens in a similar manner to the aqueous solution. In the gas phase, however, the absence of water cavity prevents the resulting fragments to reorient and react. Indeed, similar to an explosion, the two resulting fragments $·NO_2$ and $RO·$ diffuse in opposite directions, before any proton transfer can occur between them. This explains the observed direct formation of $·NO_{2,(g)}$* (Talukdar et al., 1997). *In contrast, in the cavity, collisions with the solvent are frequent, and thus, the photolysis likely follows the pathway with the minimum energy barrier, that leads to the formation of $HNO_2$ and acetone."*

**What is the role of water in these experiments? Does it help to stabilize some steps in this reaction?**

The water generates a cavity to encapsulate electrostatically and through van der Waals interactions the RONO2 molecule and its photoproducts. Besides that, we do not observe any formation of hydrogen

bonds and there is no formation of ionic species during the reaction. Therefore, the interaction with water or the effect in the electronic states is rather small.

**Is there literature precedent for this reaction in the gas or solution phase?**

To the best of our knowledge, this is the first time that this photochemical reaction has been studied theoretically.

**Minor Points.**

**As shown in Figure S5, the RONO2 compounds absorb marginally in the UV-visible portion of the actinic spectrum. Please add to the UV-visible spectra of the RONO2 compounds studied an emission spectrum of the photolysis lamp emission spectra for comparison.**

We have added Figure S1 where the lamp actinic spectra are shown as well as a comparison between the lamp spectrum and the liquid phase absorption cross-sections of the investigated $RONO_2$ reported in González-Sánchez et al., (2023).

**There are some instances where abbreviations were used without first defining them. For example, on line 72, HPIC-CD is used first here, but the abbreviation is only defined on line 113.**

Done

**I could not find a section outlining the chemical reagents and gases used, their purity, and source. Please add such a section to the Materials and Methods section.**

Done

**I notice that in discussing the time profiles of the kinetic studies that the authors refer to HNO2 and HNO3. The IC measures the anions rather than the neutral species indicated. One can discuss later how this implies that HONO can be emitted under the appropriate pH conditions; however, it is more correct to use the terms nitrite and nitrate in the text and in the legend for Figure 2b and Figure 2 caption when referring to what was actually measured.**

Yes, as exposed in the beginning of Section 3.2, both compounds were detected as $NO_2^-$, and $NO_3^-$ using HPIC-CD but their formation as acids was inferred due to the observed fast decrease of pH (Fig. S4) and was confirmed by theoretical calculations. Therefore, in the time profiles we depicted the neutral species rather than those detected. We have included in the legend of Fig. 2 a statement clarifying that HPIC-CD detected $NO_2^-$, and $NO_3^-$.

**None of the figures show any error bars. Were these experiments repeated multiple times and if so, how reproducible were they?**

For three of the investigated $RONO_2$, several experiments were conducted, i.e., isopropyl nitrate (*n*=3); isobutyl nitrate (*n*=5), and 1-nitrooxy-2-propanol (*n*=2). When performed under similar conditions (i.e., same initial concentrations), the yields were reproducible (as shown in Table S1).

**Figure 1a: In my copy of the manuscript, the x-axis label is partially missing (probably covered by a white textbox).**

Updated

**References.** In my opinion this manuscript is not consider enough of the existing literature, especially for such a complicated study done on such a well-studied topic. Also, my copy of the SI did not have any references listed although papers were cited in the text.

SI was updated with the references.

**Regarding the atmospheric significance of this work, I think the case can be made stronger if the authors refer to the literature on some of the more substituted RONO2 species that absorb more light than the model compounds they used.**

$RONO_2$ reactivity is a well-studied topic. Nevertheless, most of the literature refers to its reactivity in the gas phase, or to its kinetics. Few studies have explored the condensed-phase reactivity of individual molecules.

More substituted $RONO_2$ (e.g., carbonyl nitrates) absorbs more light, however, our previous study (González-Sánchez et al., 2023), reported similar kinetic decay for RONO2 with carbonyl groups and for alkyl nitrates. This suggests that solvent effects hinder the enhancement of their photolysis.

**References**

Brun, N., González-Sánchez, J. M., Demelas, C., Clément, J.-L., & Monod, A. (2023). A fast and efficient method for the analysis of α-dicarbonyl compounds in aqueous solutions: Development and application. Chemosphere, 137977. https://doi.org/10.1016/J.CHEMOSPHERE.2023.137977

González-Sánchez, J. M., Brun, N., Wu, J., Ravier, S., Clément, J.-L., & Monod, A. (2023). On the importance of multiphase photolysis of organic nitrates on their global atmospheric removal. Atmospheric Chemistry and Physics, 23(10), 5851–5866. https://doi.org/10.5194/ACP-23-5851-2023

Hu, K. S., Darer, A. I., & Elrod, M. J. (2011). Thermodynamics and kinetics of the hydrolysis of atmospherically relevant organonitrates and organosulfates. Atmospheric Chemistry and Physics, 11(16), 8307–8320. https://doi.org/10.5194/acp-11-8307-2011

Nguyen, T. B., Crounse, J. D., Teng, A. P., Clair, J. M. S., Paulot, F., Wolfe, G. M., & Wennberg, P. O. (2015). Rapid deposition of oxidized biogenic compounds to a temperate forest. Proceedings of the National Academy of Sciences of the United States of America, 112(5), E392–E401. https://doi.org/10.1073/pnas.1418702112

Romer Present, P. S., Zare, A., & Cohen, R. C. (2020). The changing role of organic nitrates in the removal and transport of NOx. Atmospheric Chemistry and Physics, 20(1), 267–279. https://doi.org/10.5194/acp-20-267-2020

Talukdar, R. K., Burkholder, J. B., Hunter, M., Gilles, M. K., Roberts, J. M., & Ravishankara, A. R. (1997). Atmospheric fate of several alkyl nitrates: Part 2. UV absorption cross-sections and photodissociation quantum yields. Journal of the Chemical Society - Faraday Transactions, 93(16), 2797–2805. https://doi.org/10.1039/a701781b